# Trends and Challenges in the Surveillance and Control of Avian Metapneumovirus

**DOI:** 10.3390/v15091960

**Published:** 2023-09-20

**Authors:** Gleidson Biasi Carvalho Salles, Giulia Von Tönnemann Pilati, Eduardo Correa Muniz, Antonio Junior de Lima Neto, Josias Rodrigo Vogt, Mariane Dahmer, Beatriz Pereira Savi, Dayane Azevedo Padilha, Gislaine Fongaro

**Affiliations:** 1Laboratory of Applied Virology, Department of Microbiology, Immunology and Parasitology, Federal University of Santa Catarina, Florianópolis 88040-900, Brazil; gleidson.salles@zoetis.com (G.B.C.S.); giuliavpilati@gmail.com (G.V.T.P.); marianedahmer@gmail.com (M.D.); beasavis2@gmail.com (B.P.S.); dayufsc@gmail.com (D.A.P.); 2Zoetis Industry of Veterinary Products LTDA, São Paulo 04709-111, Brazil; eduardo.muniz@zoetis.com (E.C.M.); josias.vogt@zoetis.com (J.R.V.)

**Keywords:** respiratory diseases, swollen head syndrome, *Metapneumovirus*, subtypes A and B

## Abstract

Among the respiratory pathogens of birds, the Avian Metapneumovirus (aMPV) is one of the most relevant, as it is responsible for causing infections of the upper respiratory tract and may induce respiratory syndromes. aMPV is capable of affecting the reproductive system of birds, directly impacting shell quality and decreasing egg production. Consequently, this infection can cause disorders related to animal welfare and zootechnical losses. The first cases of respiratory syndromes caused by aMPV were described in the 1970s, and today six subtypes (A, B, C, D, and two more new subtypes) have been identified and are widespread in all chicken and turkey-producing countries in the world, causing enormous economic losses for the poultry industry. Conventionally, immunological techniques are used to demonstrate aMPV infection in poultry, however, the identification of aMPV through molecular techniques helped in establishing the traceability of the virus. This review compiles data on the main aMPV subtypes present in different countries; aMPV and bacteria co-infection; vaccination against aMPV and viral selective pressure, highlighting the strategies used to prevent and control respiratory disease; and addresses tools for viral diagnosis and virus genome studies aiming at improving and streamlining pathogen detection and corroborating the development of new vaccines that can effectively protect herds, preventing viral escapes.

## 1. Introduction

Poultry farming has undergone great changes in its production systems; while the sheds where birds are housed and raised have better conditions for animal welfare and a high technological level, population densities have also increased proportionally [1], representing a large sanitary challenge. Although many measures are taken to control diseases, such as biosecurity, immunoprophylaxis, management and nutrition [2], this large population of animals in the same environment or shed becomes a risk factor for the health of the birds, which can lead to the emergence of diseases, most of which are respiratory. In this context, the flow of people, animals and migratory birds can pose a risk to the health of poultry batches [3], as this flow of people and animals can carry diseases into poultry facilities.

The blockade of respiratory diseases in birds presents a very large challenge for veterinarians, as these diseases usually do not show pathognomonic signs, that is, the clinical diagnosis is very complex [4]. Despite the presumptive diagnosis being difficult, the confirmatory diagnosis should be a common practice in the prevention of respiratory diseases. For this, a clear understanding of the means of transmission, viral incubation period, clinical signs and which material is most appropriate and when to collect it is necessary, in addition to choosing the best aMPV diagnostic method [5].

This category covers a spectrum of diseases, notably including Avian Influenza, New Castle Disease, Infectious Bronchitis of Chickens, Mycoplasmosis, Pasteurellosis, Infectious Laryngotracheitis and Avian Metapneumovirus [6].

The Avian Metapneumovirus (aMPV) is an important pathogen involved in diseases of the respiratory complex of birds, and although it is very neglected, its damage goes beyond respiratory symptoms and can affect the reproductive system [7,8,9,10,11], thus facilitating the development of other diseases, such as colibacillosis, and the association of these two diseases can cause serious damage to animals, where one disease can potentiate the effect of the other [12].

This review seeks to understand which the main subtypes are present in different countries, in addition to identifying which strategies are used to prevent and control this disease and the methods of prevention, control and viral ecology, based on the most elaborate studies to date. First, it discusses the main diagnostic tools that should be used to effectively assist in epidemiology and in the development of new tools that can protect poultry flocks, avoiding viral escapes and disease.

## 2. Characteristics of Viral Particles, Classification and Nomenclature

aMPV belongs to the Metapneumovirus genus of the *Pneumoviridae* family, it is an enveloped virus with non-segmented negative-stranded single-stranded RNA, pleomorphic spherical shape (diameters vary from 100 to 200 nm) and which can present long filaments, in addition to a helical nucleocapsid [13].

Early viral characterization through monoclonal serological assays showed some variability among aMPV strains [14,15,16,17] characterizing the molecular follow-up soon after, where genetic difference based on G protein variability was confirmed [18]. Later confirmed in several studies [19,20], this work used samples from the Central Veterinary Laboratory in Weybridge, United Kingdom. Two Subtypes, A and B, were identified, where subtype A samples came from the UK and subtype B samples from Italy and Hungary. These studies showed that the aMPV subtype A was the first to circulate in South Africa and later in the United Kingdom, and a few years later it was already possible to identify the subtype B present in Europe as well.

The aMPV genome is composed of eight viral genes (Figure 1), arranged in the order (3′-N-P-M-F-M2-SH-G-L-5′). These genes are identified as a nucleoprotein (N), phosphoprotein (P), matrix protein (M), fusion protein (F), second matrix protein (M2), small hydrophobic protein (SH), surface glycoprotein (G) and a viral (L) RNA-dependent RNA polymerase, and these genes code for nine proteins [4,21,22].

Some genes, such as G, SH, M, N, P and F, present genetic heterogeneity and can be used to differentiate subtypes. However, the G protein, a highly glycosylated type II membrane protein known for its remarkable variation in length and sequence identity even within aMPV subtypes, can also show differences in size and nucleotide number between G gene sequences of the different subtypes. Due to this greater heterogeneity in relation to other genes, based on the analysis of the nucleotide sequences of the G gene, it is possible to perform the classification and characterization of the subtypes, as well as molecular epidemiological studies [4,22].

However, a study conducted in France described isolates of subtype D (aMPV-D) exhibiting a relatively low sequence identity in the G gene compared to subtypes aMPV-A, aMPV/B and aMPV-C [4]. While a study evaluating the use of quadriplex RT-qPCR reported that the G gene can be utilized for designing primers and probes for the detection of aMPV-A, aMPV-B and aMPV-D, the M gene was used for the detection of the aMPV-C subtype [23].

The classification of different aMPV into subtypes can be supported via the RT-ddPCR technique, which was developed using a region of low variability in the L ORF, located between nucleotide positions 5′ 1980 and 5′ 2158, avoiding the limitations of previous tests. The result of this classification can be seen through the cutoff amplitude, set at ~6000 for CH1 and 3200 for CH2, the mean amplitude values for positive clusters in CH1 and CH2 were classified as aMPV samples in different subtypes. Thus, because it is based on a highly conserved region of the genome, there is less risk of the test being affected by the evolution of the viral genome and the impact it has on the specificity of such tests [24].

Protein N can be the target antigen in the development of serological assays, as it has the ability to induce a greater serological response in infected hosts. Comparison of the amino acid sequence of aMPVs indicated that within subtypes the N genes were 99–100% identical, even between viruses from different geographic regions. Amino acid identities between subtypes A and B were 90 to 91%, however, between subtypes C and A, B, or D, aa identities were only 70 to 71%, 70 to 72% and 73 to 74%, respectively [25].

## 3. Discovery and Distribution of Avian Metapneumovirus (aMPV)

The aMPV is a relatively new virus that affects turkeys and chickens and can also be found in guinea fowl [26], ducks [27] and pheasants [19]. The first report occurred in turkeys in South Africa in 1978 [28], a few years later the virus was found in chickens in England and classified as Swollen Head Syndrome (SHS) [29]. The aMPV rapidly spread across Europe, detected in the United Kingdom [30], France [31], Spain [30], Germany [32], Hungary [33], Italy [34,35], the United States [36] and Brazil [37]. Through epidemiological traceability, it was evident that the identification of subtype A was the first, however, within a few years the distribution of subtypes A and B was already identified in other countries, making control difficult [18].

In the early 1980s, the transmission between different species of birds was still unclear, but in 1987, through an experiment carried out by Picault et al., 1987, where they isolated a virus in chickens that became ill with aMPV; this virus was removed, homogenized from the respiratory tract tissue and was later inoculated into SPF turkeys and these showed characteristic clinical signs of avian Metapneumovirus, thus evidencing the transmission between chickens and turkeys [38].

The aMPV has a wide global distribution, essentially where there are poultry production or migratory bird routes the virus can be identified, but what changes is the prevalence of subtypes A, B, C and D, in addition to the two new subtypes described: (a) subtypes A and B are likely to be found in Europe, Brazil and the African continent [39]; (b) the C subtype has been identified in the United States, Canada, China, France and recently in South Korea [40,41,42,43,44,45]; and (c) the D subtype was only reported in France [39] and the two new subtypes were found in the United States and Canada [45]. Today, the most prevalent subtype in the world is B [22,46].

Changes related to the G protein have an impact not only on subtyping characteristics, but also on virus replication in the target cell, thus changing its pathogenicity in the host [22].

## 4. Replication, Viral Persistence and Clinical Signs

The intense replication of the virus in the upper tract of birds (sinuses, larynx and trachea) causes the cessation of ciliary movements (ciliostasis), which can lead to the complete loss of these cilia (desciliation) [47]. This process results in difficulty in removing mucus, which accumulates in the passages and cavities, and gives rise to the main clinical sign of the disease, the swollen head, although this condition is not always present [22]. This primary infection favors the invasion of secondary agents, such as *E. coli,* which cause different clinical signs and whose intensity is linked to the pathogenicity of the agents involved. The aMPV is the cause of severe respiratory infection in turkeys, Turkey Rhinotracheitis (TRT), and usually occurs in young birds [48,49].

Common symptoms include sneezing, nasal and eye discharge, conjunctivitis, submandibular edema, infraorbital sinus swelling, cracking and rales [48,49,50]. In chickens, the virus has been associated with Swollen Head Syndrome (SHS), which is characterized by swelling of the periorbital and infraorbital sinuses, torticollis, disorientation and opisthotonos [50]. The clinical manifestation may progress to redness of the conjunctiva with edema of the lacrimal gland. After 12 to 24 h, the birds show a subcutaneous swelling on the head, which starts around the eyes, increases under the entire head and descends to the submandibular tissue and back of the neck. After three days, they may show neurological signs such as apathy and torticollis [48,49].

The permanence period of aMPV is extremely short in birds, not exceeding 4 to 7 days, which greatly impairs virus detection for molecular diagnosis [16,50].

## 5. Transmission and Economic Losses

The most common route of transmission of aMPV occurs horizontally through aerosol however, there are other paths taken by the virus until contact with birds, such as water, equipment, feed trucks that supply the farms and the transit of people [51].

So far, there is no clear evidence of vertical contamination through breeders to progeny [48,52,53]. In addition, migratory birds play an important role in the spread of the virus; there are reports of outbreaks of clinical cases of aMPV in birds in periods that coincide with the migration of wild birds [41]. The detection of antibodies in geese, house sparrows, gulls, parakeets, waterfowl and several other species suggests the circulation of this virus by wild birds [44,45,47,51], which reinforces the need for serological monitoring of these animals.

Economic losses in broilers due to respiratory complications related to aMPV alone or with secondary bacterial infections affect 1% to 3%, and 20% to 30% of cases, respectively [49]. In commercial batches, the first signs are mild respiratory failure, rhinitis and conjunctivitis, followed by neurological signs and swollen head. Reproductive alterations can be observed and alterations in the production or quality of the eggs can be common [7,50,54,55].

Morbidity and mortality are influenced by co-infections. When chickens show clinical signs, morbidity at all ages is often described as up to 100%, as mortality ranges from 0.4% to 50%, particularly in susceptible young birds [49].

## 6. Co-Infection of aMPV and Bacteria

Cases of aMPV often coincide with co-infection with *Escherichia coli* [7,48,49,50]. When such co-infection occurs, the clinical signs in birds tend to be more severe. This is because one agent can potentiate the action of the other, thereby increasing the overall pathogenicity of the clinical presentation [51,52,53,54,55,56,57].

Studies have shown high morbidity and exacerbation of the clinical picture in turkeys co-infected with aMPV and *Mycoplasma gallisepticum* [58], *Ornithobacterium rhinotracheale* [59] and lentogenic Newcastle disease virus [60]. Chickens experimentally infected with aMPV and later infected with three different bacteria (*Escherichia coli*, *Bordetella avium*, *Ornithobacterium rhinotracheale* or a mixture of the three) were evaluated; animals infected with aMPV and the mixture of the three developed more severe clinical symptoms when compared to birds inoculated with aMPV or bacteria alone. The air sacs and lungs in this situation showed more severe alterations in birds inoculated with aMPV and *Bordetella avium* [61].

Infections by aMPV and *Ornithobacterium rhinotracheale* are one of the main problems related to the respiratory system in turkeys. Field cases and experimental studies have shown that the most common manifestations in cases of co-infection are airsacculitis and pneumonia [62,63,64].

## 7. Vaccination against aMPV and Viral Selective Pressure

The first vaccine used to control and prevent aMPV originated from a field strain called UK/3B/85 belonging to subtype A [65,66]. Until 1995 only vaccines belonging to subtype A were used in the UK, and the prevalence of this subtype appears to have declined, in contrast, what has been observed is an increase in the prevalence of aMPV subtype B [67].

This possible selection by vaccine pressure was demonstrated in a study in which eight aMPV strains (pre-1994) and six aMPV strains were evaluated between 2001 and 2007, with these samples coming from the Veneto region, Italy.

The strains, when compared, showed genetic mutations in specific amino acids of glycoprotein G, and the prevalence was of aMPV subtype B and the vaccines used also belonged to this same subtype [11]. These genetic mutations in glycoprotein G may favor viral escapes, as vaccines will present partial coverage, which would reduce their effectiveness when used.

Although vaccines played an excellent role in controlling aMPV with a homologous subtype, the pressure exerted by vaccines on the environment may have helped in the dominance of another subtype, in this case B. In addition to vaccine pressure, hosts and environment can help in this process. In studies [68,69,70,71], the heterologous protection capacity between vaccines of different subtypes is demonstrated, although it is susceptible to viral escapes.

There are two types of live vaccines available on the market, one subtype A and the other subtype B, information in the literature indicates that both products provide good cross-protection [68,69,70,71].

Other very important measures, such as a rigorous biosecurity program and proper management, including bird density, litter conditions, sanitary intervals, cleaning and disinfection, multiple ages and environmental conditions (ventilation, temperature variations) are also of great importance for the successful control of this disease [72]. Thus, in high-risk regions, vaccination together with a biosecurity program is an indispensable part of the strategic control of aMPV. The introduction of vaccines into immunoprophylaxis programs is not a simple activity, as in addition to the costs involved, there is an enormous limitation of labor for the application of vaccines, whether through mass (spray, drinking water) or individual (ocular or intramuscular) application, which require greater attention, as in these cases vaccines are applied bird by bird, which often limits the use of this method. [73].

Vaccination programs can be used through different strategies, it is important to know how they work and what the final objective is of protection for aMPV. It is possible to use replicating (live) and inactivated (non-replicating) vaccines, and it is important to respect the purpose of each tool. Replicating vaccines end up stimulating both a cellular and humoral response. When necessary, these vaccines can be used in broiler chickens from the first day of life. Non-replicating vaccines are widely used in long-lived birds (broiler-breeders). Non-replicating vaccines induce greater production of circulating antibodies (IgY) mainly to protect the reproductive tract of birds in addition to reducing viral excretion [74]. In long-lived birds, the ideal would be to associate the two technologies, where the replicating vaccine would serve as a primer for the non-replicating vaccine, providing broader and more uniform protection for birds [75].

aMPV has been considered a relatively slow-evolving virus when compared to other avian RNA viruses, however, other studies estimate that this rate of viral evolution is within the normal range [76,77,78]. Viral evolution is based both on the pressure exerted by vaccine programs and on the type of host and the environment, since different strains belonging to the same subtype circulate phenotypically in different regions of the world [78].

Thus, the vaccines used for the prevention and control of aMPV, although capable of reducing clinical conditions and viral dissemination, are not equally effective in preventing aMPV infection and circulation, contributing to viral persistence within a given region or country [78].

It is very important to monitor the animals, this includes indirect methods such as serology, but molecular diagnosis is essential to identify the prevalence of the subtype, which can help in making more assertive decisions [79,80].

## 8. Methodological Trends for the Discovery of New Viral Strains in Poultry

Molecular techniques such as PCR or RT-qPCR, Sanger sequencing and next generation sequencing (NGS), have demonstrated their clinical and epidemiological importance through the identification and genetic typing of pathogens [80,81,82,83]. The NGS and Sanger sequencing techniques are distinguished by their ability to identify mutations during the sequencing process, where Sanger is used to identify short DNA sequences (500 to 900 base pairs), while NGS is capable of sequencing 50 to 900 base pairs 300 nucleotides in length, in addition to its ability to read billions of genetic fragments at the same time [84]. While the Sanger sequencing technique is confined, in clinical practice, to the detection of point mutations, NGS has introduced significant innovations. Although NGS does have some limitations related to the potential for errors in genomic regions with repetitive nucleotide bases, it stands out for its enhanced capability to analyze mutations during the same process, as well as its greater efficiency and speed in generating and identifying results [81,82,83].

NGS techniques seem more accurate in identifying not only point mutations, but also already circulating or new variants (whole genomes) of infectious and contagious pathogens, such as aMPV [83,84]. NGS is characterized via DNA or cDNA sequencing, which can generate short or long reading fragments, depending on the methodology used. The most used NGS platforms for respiratory virus detection are: Illumina sequencers, Life Technologies sequencers, Oxford Nanopore sequencers and Roche (Metapneumovirus) sequencers [84,85].

Considering that aMPV is one of the most important respiratory agents in birds and associated with economic losses in production, and because it is an RNA virus, where mutations and recombinations occur at higher rates, genomic surveillance is an important tool for tracking the dissemination of variants and monitoring of genetic alterations [84,85,86,87,88,89].

Currently, serological methods are widely used for screening and field monitoring of batches where serum antibody titers are detected. However, molecular techniques, mainly RT-qPCR, are widely used for aMPV detection, where primers are usually designed to amplify the G gene region both for viral detection and for the identification of subtypes, as it is a region of substantial heterogeneity [88].

In Brazil, there are studies describing the circulation of the two main subtypes of aMPV, A and B. However, it is important to highlight that the epidemiological survey of the pathogen is outdated, as there is no genomic surveillance system implemented in the country. Therefore, whole-genome sequencing techniques, such as next-generation sequencing (NGS), are essential for the identification of small mutations in the aMPV genome, which may lead to vaccine inefficiency, as well as for the development of vaccines [87,88,89,90].

Kariithi et al. (2022) used the Illumina MiSeq platform to sequence complete genomes of aMPV subtype A in broilers from Mexico [90]. Based on the recent impact caused by aMPV at the clinical and economic levels in Europe, the platform reconstructed the phylogeny and viral dispersion based on sequences deposited in GenBank from 1985 to 2019 and identified the heterogeneity of circulating strains among the countries analyzed and, although the authors did not report any significant host adaptation, there was a shift in bloodlines between turkeys, guinea fowls and chickens; this heterogeneity can lead to low coverage and vaccine failures [22].

Unfortunately, despite the visible need for more accurate genomic surveillance, genetic analysis of aMPV is still very scarce. Currently, studies with SNG are more focused on human Metapneumovirus, with birds being the majority analyzed via RT-PCR (Table 1).

The studies presented here affirm the importance of effective genomic surveillance, including for aMPV, which can influence prevention, clinical and economic improvement in animals for producers, as well as improvements in animal diagnosis and therapy and the development of more effective vaccines that have greater protection coverage compared to current aMPV vaccines.

## 9. Conclusions

The evolution of molecular techniques for viral diagnosis, mainly of aMPV, has played an important role in fast and accurate viral detection, understanding the epidemiology and helping to inform the best control strategies over the years. Although there are important tools to support and assist in accurate diagnosis, it is essential to evolve further in the detection and control of this disease. In this sense, in addition to traditional serological and molecular methods, we emphasize genomic sequencing which has allowed the broad characterization of the genetic variability of the virus, the detection of mutations, and the identification of new variants, as well as allowing the simultaneous evaluation of pathogens present in the same sample in cases of co-infections. This approach has allowed us to understand the evolution of viruses over time and can be used to evaluate the efficiency of vaccination programs in poultry.

It is evident that the control of aMPV requires a holistic view, focused on knowledge of the agent, epidemiological surveillance, effective diagnosis, adequate immunoprophylactic programs and constant discussing of the precepts of biosafety, thus, a new approach must be used for control and disease prevention, mainly respiratory, in birds.

## Figures and Tables

**Figure 1 viruses-15-01960-f001:**
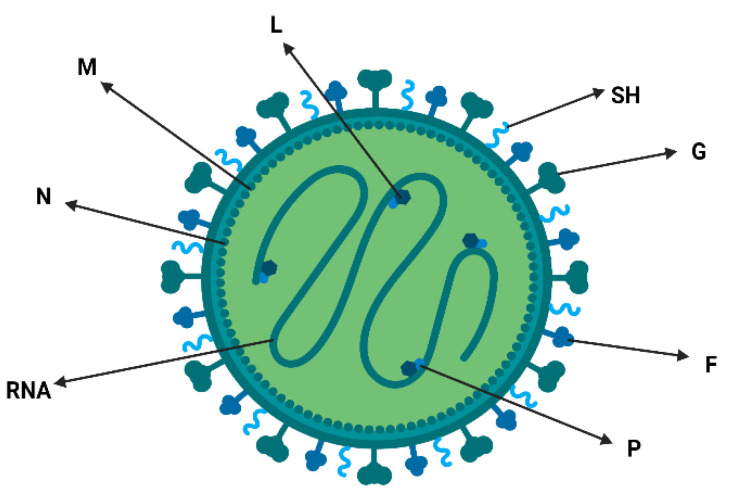
Schematic figure representing aMPV: (G) Glycoprotein, (F) Fusion protein, (SH) Small hydrophobic protein and other structural proteins, (M) Matrix protein, (N) Nucleocapsid protein, (P) Phosphoprotein, (L) RNA-dependent RNA polymerase and RNA strand.

**Table 1 viruses-15-01960-t001:** Epidemiological analysis studies based on aMPV genomic analysis.

Target	Sample	Methodology	Country	aMPV Subtype	Publication Year	Author
Gene—G	Cloacal/throat double swabs	RT-PCR	China		2022	[91]
Gene—G and Protein G	Choanal cleft swab	ELISA	North Vietnam		2021	[92]
		RT-PCR		aMPV B	2021	
Genes—G, N e M	Choanal cleft swab	RT-PCR	Iran	aMPV B	2017	[93]
Gene—G	Respiratory tract swabs	RT-PCR	Northern Italy	aMPV B	2018	[86]
Gene—M	Oropharyngeal and cloacal swabs	RT-PCR	Canada	aMPV C	2018	[45]
Gene—G	Tracheal swabs	RT-PCR and Sanger sequencing	Greece	aMPV B	2019	[94]
Gene—G	Throat swabs	-	China	aMPV B	2019	[95]
Gene—N, M, F, L, M2, SH e G	Tissues swabbed (choana, lung)	Illumina sequencing	Mexico	aMPV A	2022	[96]
Genome	Uninformed	NGS (Illumina MiSeq)	Hungary	aMPV B	2020	[97]
Gene—G	Tracheal and cloacal swabs	RT-PCR and Sanger sequencing	Brazil	aMPV A and B	2011	[72]

## Data Availability

Not applicable.

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
