# Peer review of "Trends and Challenges in the Surveillance and Control of Avian Metapneumovirus"

_viruses, 2023, doi:10.3390/v15091960_

Round 1
Reviewer 1 Report (Previous Reviewer 2)
Could the Authors describe better the difference limits for considering the samples belonging to different subtypes: for examples, the G gene should present a limit of 10% (or 5%, or 15%) in nucleotide difference for considering a different subtype... or it considers the aminoacidic level?
table1. the "Country" names are not corrected (Japao...)... check all the manuscript...
Author Response
We appreciate the review.
The text has been extensively revised for grammar and content details. Below we highlight the main revisions included in the text (view MS in yellow color).
# Comments and Suggestions for Authors / R1
Could the Authors describe better the difference limits for considering the samples belonging to different subtypes: for examples, the G gene should present a limit of 10% (or 5%, or 15%) in nucleotide difference for considering a different subtype... or it considers the aminoacidic level?
Response:
We agree with the reviewer's suggestion.
1 -These points have been clarified and the following paragraphs have been edited:
“ Some genes, such as G, SH, M, N, P and F, present genetic heterogeneity and can be used to differentiate subtypes. However, G protein, a highly glycosylated type II membrane protein known for its remarkable variation in length and sequence identity even within aMPV subtypes, can also show differences in size and nucleotide number between G gene sequences of the different subtypes. Due to this greater heterogeneity in relation to other genes, based on the analysis of the nucleotide sequences of the G gene it is possible to perform the classification and characterization of the subtypes, as well as molecular epidemiological studies [4,22].
However, a study conducted in France described that isolates of subtype D (aMPV-D) exhibited a relatively low sequence identity in the G gene compared to subtypes aMPV-A, aMPV/B, and aMPV-C [4]. While a study evaluating the use RT-qPCR reported that the G gene can be utilized for designing primers and probes for the detection of aMPV-A, aMPV-B, and aMPV-D, the M gene was used for the detection of the aMPV-C subtype [23].
The classification of different aMPV into subtypes can be supported by the RT-ddPCR technique, which was developed using a region of low variability in the L ORF, located between nucleotide positions 5' 1,980 and 5' 2,158, avoiding the limitations of previous tests. The result of this classification can be seen through the cutoff amplitude, set at ~6,000 for CH1 and 3,200 for CH2, the mean amplitude values ​​for positive clusters in CH1 and CH2 classified as aMPV samples in different subtypes. Thus, because it is based on a highly conserved region of the genome, there is less risk of being affected by the evolution of the viral genome and the impact it has on the specificity of such tests.”
2 - Table1. the "Country" names are not corrected (Japao...)... check all the manuscript...
Response: Thank you for the observation. Table 1 has been corrected.
Reviewer 2 Report (New Reviewer)
The article entitled “Trends and challenges in the surveillance and control of Avian Metapneumovirus by Salles et al. is an interesting short review. however, it requires minor improvements to ensure its quality and increase its reader range.
1. Line 48: Please put "Avian Metapneumovirus (aMPV) " and then use the abbreviation "aMPV" throughout the MS
2. line 67: Pneumoviridae should be italic
3. Line 164: please elaborate more on O. rhinotracheale as it is one of the most critical problems please refer to the following “Hafez, H. M. and Shehata, A. A. 2021. Turkey production and health: current challenges. doi: 10.51585/gjvr.2021.0002 “, “van Empel, P.C., Hafez, H.M., 1999. Ornithobacterium rhinotracheale: A review. doi: 10.1080/03079459994704”
4. Section 7. Vaccination against aMPV and viral selective pressure, I would suggest adding a short paragraph about the suitable age of vaccination, is it applicable in broilers or not supported with references?
5. Line 205, please elaborate on the needed manpower, is it related to the application for individual birds i.e. by eyedrop for example?
6. Sections 7. Vaccination against aMPV …. & 8. Methodological trends for the discovery …. are too short paragraphs that need to be improved in terms of linking between this information. These short paragraphs are just short glances at each piece of information., Please improve
7. Section 8. Methodological trends for the discovery ….: I would suggest a thorough explanation of the advantages of NGS if any, and justification for the need for NGS if the conventional Sanger method is doing its job.
8. line 253, the sentence started with "(2022) used the…", please correct
9. Table 1: Please include the main finding of these references, e.g. what subtype, and are these the only available studies, to ensure a wide range of readers, please include more countries, if applicable
none
Author Response
We appreciate the review.
The text has been extensively revised for grammar and content details. Below we highlight the main revisions included in the text (view MS in yellow color).
# Comments and Suggestions for Authors / R2
The article entitled “Trends and challenges in the surveillance and control of Avian Metapneumovirus by Salles et al. is an interesting short review. however, it requires minor improvements to ensure its quality and increase its reader range.
- Line 48: Please put "Avian Metapneumovirus (aMPV) " and then use the abbreviation "aMPV" throughout the MS
Response: Thank you for the suggestion, the abbreviation has been added throughout the manuscript.
- line 67: Pneumoviridae should be italic
Response: Thank you for the correction, the word has been corrected and italicized.
- Line 164: please elaborate more on O. rhinotracheale as it is one of the most critical problems please refer to the following “Hafez, H. M. and Shehata, A. A. 2021. Turkey production and health: current challenges. doi: 10.51585/gjvr.2021.0002 “, “van Empel, P.C., Hafez, H.M., 1999. Ornithobacterium rhinotracheale: A review. doi: 10.1080/03079459994704”
Response: Thank you for the suggestion, a paragraph discussing co-infection was added based on requested articles.
“Infections by aMPV and Ornithobacterium rhinotracheale are one of the main problems in turkeys related to the respiratory system. Field cases and experimental studies have shown that the most common manifestations in cases of co-infection are airsacculitis and pneumonia.”
- Section 7. Vaccination against aMPV and viral selective pressure, I would suggest adding a short paragraph about the suitable age of vaccination, is it applicable in broilers or not supported with references?
Response: Thanks for the suggestion, we've added a discussion paragraph.
“The introduction of vaccines into immunoprophylaxis programs is not a simple activity, as in addition to the costs involved, there is an enormous limitation of labor for the application of vaccines, whether through mass (spray, drinking water) or individual (ocular or intramuscular), which require greater attention, in these cases, vaccines are applied bird by bird, which often limits the use of this method. [74].
Vaccination programs can be used through different strategies, it is important to know and know how they work and what is the final objective of protection for aMPV. It is possible to use replicating (live) and inactivated (non-replicating) vaccines, it is important to respect the purpose of each tool. Replicating vaccines end up stimulating both a cellular and humoral response, when necessary, these vaccines can be used in broiler chickens, from the first day of life. Non-replicating vaccines are widely used in long-lived birds (broiler-breeders). Non-replicating vaccines induce greater production of circulating antibodies (IgY) mainly protect the reproductive tract of birds in addition to reducing viral excretion [75]. In long-lived birds, the ideal would be to associate the two technologies, where the replicating vaccine would serve as a primer for the non-replicating vaccine, providing broader and more uniform protection for birds [76].
- Line 205, please elaborate on the needed manpower, is it related to the application for individual birds i.e. by eyedrop for example?
Response: Thanks for the suggestion, changes have been made to the paragraph.
“The introduction of vaccines into immunoprophylaxis programs is not a simple activity, as in addition to the costs involved, there is an enormous limitation of labor for the application of vaccines, whether through mass (spray, drinking water) or individual (ocular or intramuscular), which require greater attention, in these cases, vaccines are applied bird by bird, which often limits the use of this method. [69].”
- Sections 7. Vaccination against aMPV …. & 8. Methodological trends for the discovery …. are too short paragraphs that need to be improved in terms of linking between this information. These short paragraphs are just short glances at each piece of information., Please improve
Response: Thanks for the note. The paragraphs were complemented and improved and are underlined throughout the text in topics 7 and 8.
“Considering that aMPV is one of the most important respiratory agents in birds and associated with economic losses in production, and because it is an RNA virus, where mutations and recombinations occur at higher rates, genomic surveillance is an important tool for tracking the dissemination of variants and monitoring of genetic alterations [80,81,82].”
“Currently, serological methods are widely used for screening and field monitoring of batches, where serum antibody titers are detected. However, through serology it is not possible to identify circulating subtypes. On the other hand, molecular techniques, mainly RT-qPCR, are widely used for aMPV detection, where the primers normally used are the G gene sequence for viral detection and identification of subtypes, as it is a region of substantial heterogeneity [83].”
“In Brazil, there are studies describing the circulation of two main subtypes of aMPV, A and B. However, it is important to highlight that the epidemiological survey of the pathogen is outdated, as there is no genomic surveillance system implemented in the country [84]. Therefore, whole-genome sequencing techniques, such as next-generation sequencing (NGS), are essential for the identification of small mutations in the aMPV genome, which may lead to vaccine inefficiency, as well as for the development of vaccines. a genomic surveillance system for the virus [84,85].”
- Section 8. Methodological trends for the discovery ….: I would suggest a thorough explanation of the advantages of NGS if any, and justification for the need for NGS if the conventional Sanger method is doing its job.
Thank you for the suggestion. The paragraph was complemented and improved.
“While the Sanger sequencing technique is confined, in clinical practice, to the detection of point mutations, Next-Generation Sequencing (NGS) has introduced significant in-novations. Although NGS does have some limitations related to the potential for errors in genomic regions with repetitive nucleotide bases, it stands out for its enhanced ca-pability to analyze mutations during the same process, as well as its greater efficiency and speed in generating and identifying results”
- line 253, the sentence started with "(2022) used the…", please correct
Thank you for the suggestion, the term has been corrected in the manuscript.
- Table 1: Please include the main finding of these references, e.g. what subtype, and are these the only available studies, to ensure a wide range of readers, please include more countries, if applicable
Thank you for the suggestion. The subtypes found were described in Table 1.
This manuscript is a resubmission of an earlier submission. The following is a list of the peer review reports and author responses from that submission.
Round 1
Reviewer 1 Report
The manuscript "Trends and challenges in the surveillance and control of Avian Metapneumovirus" is a review focusing on the description on various aspects of the virus, leading to highlighting the need of a stricter surveillance and the application of control measures.
However, the manuscript is prepared in a superficial way, the contents are listed but not deepened, the references lead sometimes to other reviews and the paper does not cite the original article supporting the concept (line 82, ref 4 and 22).
Many sentences are incomplete or unclear (lines 15, 38-40, 63-66, 72, 79-81, 92-96, 97-99, 211-213, ...), some sencences or periods are redundant or not logically organized (lines 38-47), the names of the pathogens are not in italics (lines 59-61, 147-159), the table is not in english.
Due to the nature of the paper, a review paper should be more well-finished, with a better use of references pointing at the direct source of information and it should follow a more thorough logical thread, using and organizing published information to suggest and support original and useful considerations.
English writing should be checked along the manuscript, together with wording and the structure of some sentences, which does not seem very english (lines 39-40, 98, 167).
Reviewer 2 Report
the Authors generally describe what it's known about aMPV in this review.
General comments:
the "epidemiological" comment is not so clear:
pag 3: "based on the nucleotide sequences...". What kind of method is used to divide aMPV into subtype? nucleotide difference? of which gene?
pag 6: about NGS, what are the most variable genes? the G one? what are the heterogeneity found in this papers? is the genetic variability important for the diagnostic success?
pag 6: It should be better to give a more complete overview about the target of each test, both serological and molecular ones, reported in table 2. In this way the information will be more complete. In the same way, the information about the target for the tests (gene or protein) should be added to the text.
pag 7: the conclusions are too short... please increase the section, with comments on tests' validity and peculiarity.
Specific comments:
Table 2. be careful to write/traslate all its part in English: "NGS nao visado" is not!
Other comments:
Do you have conflict of interest? Four authors have Zoetis affiliation... no problem, but they have to declare it...
In the pdf file there
the English is acceptable (except the untranslated part...)